# Revising the Environmental Kuznets Curve for Deforestation: An Empirical Study for Bulgaria

**Stavros Tsiantikoudis [1,\*], Eleni Zafeiriou [2], Grigorios Kyriakopoulos [3]**  **and Garyfallos Arabatzis [1]**

[1]   Department of Forestry and Management of the Environment and Natural Resources, Democritus University of Thrace, GR68200 Orestiada, Greece
[2]   Department of Agricultural Development, Democritus University of Thrace, GR68200 Orestiada, Greece
[3]   School of Electrical and Computer Engineering, National Technical University of Athens (NTUA), Division of Electric Power, GR15780 Athens, Greece
**\***   Correspondence: stsianti@fmenr.duth.gr

**Abstract:** The evolution of human societies along with efforts to enhance economic welfare may well lead to the deterioration of the environment. Deforestation is a usual process throughout evolution that poses pressing and potentially irreversible environmental risks, despite the ecological and modernization processes that aim to limit those risks. The economic growth–environmental degradation relationship—namely, the environmental Kuznets curve (EKC) hypothesis—is studied in alignment with the autoregressive distributed lag (ARDL) approach. The novelty of the study is attributed to the use of the carbon emissions equivalent derived by deforestation as an index for environmental degradation in Bulgaria as a new entrant into the European Union (EU). In addition, we use the gross domestic product (GDP) per capita as a proxy for income, being determined as an independent variable. Research findings cannot validate the inverted U-shape of the EKC hypothesis; instead, an inverted N pattern is confirmed. The implementation of appropriate policies aiming at the protection of the environment through the diversification of economic activities is related to the use of forest land and other resources, or related sectors (agroforestry, ecotourism activities, and scientific research), rather than only the direct utilization of forested areas; the limitation of afforestation processes and their negative impacts on citizens' welfare are also addressed.

**Keywords:** environmental Kuznets curve; deforestation; ARDL with bounds test

## 1. Introduction

The multifunctionality of forests in Europe stems from the diversity of tree species and ecosystems. Indicatively, forests serve as a regulatory tool for hydrologic cycles, provide refuges for biodiversity, contribute raw material for medicinal and forest products, prevent soil destruction, and satisfy recreational, spiritual, and aesthetic value needs [1]. However, the most significant impact is related to climate change mitigation, since it contributes greatly to exchanges in energy, water, carbon dioxide, and other chemical substances [1,2]. The significant contribution of forests to climate change mitigation has become recognized in the existing literature recently [3]. Given that the destruction of forests reduces the ability of the Earth to absorb $CO_2$ from the atmosphere [4], an effort was initiated for its limitation. This effort was established with the negotiation of the Montreal UNFCCC CoP held in 2005, in which the dominant motto for forest management is "Reducing emissions from deforestation and forest degradation" (REDD).

In Europe, forests correspond to 37% of the terrestrial surface with ecological, economical, and soci-ocultural impacts, as mentioned above [5,6], while eastern and central European countries do still have large and relatively undisturbed forests compared to western European countries [7,8].

Within the last couple of centuries in the name of societal modernization and urbanization, deforestation has become a necessary tool, which permanently converts forest land into other land uses [9].

A high level of deforestation is globally considered an anthropogenic environmental problem [9,10]. The impacts of this problem are more severe for less developed countries, and less extended for developed countries. Indeed, developed countries are exerting intense efforts for afforestation, which led to an increase in the forest area, estimated at 1% from 1990 to 2005 [11]. The linkage among deforestation, forest degradation, and climate change is attributed to fewer trees absorbing less greenhouse gases (GHGs); therefore, deforestation is indirectly causing increased quantities of carbon dioxide emissions [12,13]. In addition, according to macro-sociologists, carbon dioxide emissions may be indicative of marketable outsourcing production for developing countries [13]. Deforestation in global terms is related to the cooling effect and the warming carbon cycle, due to changes in albedo and evapotranspiration [14].

Deforestation in the European Union (EU) constitutes an environmental issue of adverse negative impacts within the last few decades. Motivation policies for deforestation are characterized by complexity and are differentiated from region to region and from country to country. One of the major reasons for deforestation in the EU, according to the FAO (2016) [15], is related to the agricultural expansion for the production of specific food commodities (accounting for 80%), while the issues of urbanization/infrastructure interpret less than 10% of deforestation each.

Bulgaria is selected as the country to be studied, since it has become a new member state in the EU in 2007 and it is ranked fourth among EU members in terms of gross domestic product (GDP) growth for the last decade. The change from a highly centralized, planned economy to an open, market-based, upper-middle income economy has become achievable after a decade of slow economic restructuring and growth, high indebtedness, and a loss of savings. The time period until its entrance into the EU was characterized by an exceptionally high economic growth and improved living standards. However, in the last decade, few positive impacts were implemented. Particularly, the global economic crisis of 2008 and a period of political instability in the period 2013–2014 urged the Bulgarian government to achieve the objectives of growth and shared prosperity.

Today, for Bulgaria, the most importance issues to be addressed are those of raising productivity and handling the rapid demographic change. Particularly, higher productivity growth is critical to accelerate convergence, as Bulgaria's income per capita is only 47% of the EU average, which is the lowest in the EU.

Furthermore, in Bulgaria, according to statistics provided by the FAO [15], 36.1% or about 3,927,000 ha of the total area is forested, 8.6% (338,000.00) of which is classified as primary forest, which is the mostly bioversatile and carbon-dense forest area. In addition, 815,000 ha are forest plantations. Besides, deforestation has currently become a significant issue, since Bulgaria's forests contain 202 million metric tons of carbon in living forest biomass.

The overexploitation of forests—being not accompanied by a project for sustainable management—as well as a lack of effective surveillance of forests and forest areas in Bulgaria resulted in losing an average of 30,000 ha or 0.90% per year [16].

Common methods of deforestation for Bulgaria are burning trees and clear cutting. These practices are considered controversial, since they leave the land completely barren. Degradation and deforestation may well lead to a devastating chain of events both locally and globally, including the loss of species, the water cycle (trees are important to the water cycle, since they absorb rain fall and produce water vapor that is released into the atmosphere), soil erosion, life quality, and floods during winter.

The low income is encouraging the government to implement institutional and legislative changes, such as low tax rates and promotional investments. On the other hand, the country's forest cover

has been expanding gradually, at an annual rate of 0.6% over the period 1990–2015 [17], while as an EU member, Bulgaria is required to implement the EU Timber Regulation, which came into force in March 2013.

Another major alarming problem with the forests in Bulgaria is illegal logging. Among the types of illegal logging are the illegal lending of forest area and damage of forest stands to obtain extended harvesting volume at a lower price.

Nowadays, the legislation has become strict regarding the activities and rights related to forest management and timber harvesting. The state forestry staff should control logging activities, but all the violations in forests are caused by a lack of governance and enforcement of existing regulations and laws, thus necessitating cooperation among forest employees. Moreover, low salaries have led to high rates of corruption in terms of state forest services as well as to limited and ineffective control mechanisms [16]. It is noteworthy that deforestation and forest degradation account for approximately 20–25% of global anthropogenic greenhouse gas (GHG) emissions, which are the major source of emissions from developing countries [18]. Deforestation in Bulgaria, as among all ex-socialist countries, and especially in private forests, is becoming limited according to statistical data, which is attributed to the modernization, urbanization, and immigration phenomena that were initiated in the period of socialism. In Bulgaria, data are available only for the case of afforestation, but not for deforestation. An indirect conclusion can be reached through the change in the land area covered by forests as well as the afforestation rate. Particularly, a decrease in the afforestation rate was observed during the 1990s, when the afforested areas per year were decreased under 10,000 ha year$^{-1}$, and after 2009, under 5000 ha year$^{-1}$. The major reasons for this decrease are the following: the large mastered areas for afforestation, the development of silvicultural systems with natural regeneration priority, chronic economic crises, as well as a lack of resources.

There are plentiful studies addressing the environmental Kuznets curve (EKC) hypothesis, where deforestation has been used as a proxy for environmental degradation, given the direct effects of growth on natural capital, in order productive agrarian efforts and policies to be promoted [19]. A lack of data for the deforestation area was the reason that the authors in this study used the GHG emissions generated by deforestation as an index for environmental degradation.

In this research context, the extended deforestation and the carbon emissions generated by deforestation are related to the GDP per capita (as motivation for income improvement). This scientific background can be a subject of econometric study under the framework of the EKC on providing policy tools to enable a strategy design that can provide alternative and more profitable sources of income. In this study, the ARDL bounds cointegration technique was deployed, and it validated the reversed N Kuznets pattern for the data applied. The novelty of the study stands on the use of the GHG emissions generated by deforestation for a country with many particularities regarding the issue of deforestation, including a lack of data, suffering from the problem of illegal logging, and strong motivation for afforestation. The study is organized as follows: Section 2 describes the existing literature, Section 3 outlines the methodology, Section 4 provides and discusses the results, and the concluding remarks are succinctly presented in Section 5.

## 2. Literature Overview

### 2.1. Environmental Kuznets Curve Hypothesis (EKC)

The EKC hypothesis is an empirical relationship that assumes the existence of a relationship between environmental quality or pollutant emissions and economic growth. Two plausible explanations have been suggested for the interpretation of the environmental–economic performance relationship [20,21]. The first interpretation is related to an income effect, because the environment is valued as a luxurious asset. Therefore, in the initial stages of the economic development process, the individuals are not willing to trade consumption for investment in environmental protection, resulting in a decline in environmental quality. Once individuals reach a specific level of consumption, which is

known in the EKC literature as the "income turning point", they ask for increasing investments for improving the environment. In addition, after the turning point, environmental quality indicators (showing pollution and environmental degradation) begin to improve.

Potential factors for the interpretation of the EKC are the following: first, the scale of production related to production expansion with the mix of products produced, and the mix of production inputs used, is considered constant under the condition of technological status. Second, different industries are characterized by different pollution intensities, while the output mix typically changes along with economic development. Third, changes in the input mix may well be related to the substitution of less environmentally damaging inputs to production for more damaging inputs, and vice versa. Another significant factor is the trade openness and specifically the increasing trade openness in the case of Tunisia, which is validated as statistically significant with both linear and nonlinear ARDL [22]. Four, it is noteworthy that the improvements in the state of technology, especially those changes in production efficiency and emissions-specific changes in process that result in less pollutant emitted per unit of input.

Nevertheless, there has been not a consensus on the validation or rejection of the EKC hypothesis. Specifically, several studies are devoted to environmental degradation as an endogenous variable and income per capita as an exogenous variable with mixed results [23].

*2.2. Previous Studies*

2.2.1. EKC General

Starting from Grossman and Krueger [24], plentiful studies can be mentioned with differences in terms of study period, methodological adaptation, power of income, and choice of control variables [25]. The data employed can be time series cross-section, or even panel data, also with different results [26,27]. For the time period 1991–2009, the time-series analysis involved nearly eight broad categories of methods, while the results obtained from these studies are inconclusive. An inverted U-shape was validated for France by Ang [28] for 1960–2000 with the assistance of ARDL methodology; Ozatac et al. also validated an EKC pattern for Turkey [29] with the same methodology for the period from 1960 to 2013. On the other hand, the N-shape EKC pattern was a finding by Akbostancı et al. [30] for Turkey for the period from 1968 to 2003, with the assistance of the cointegration technique. Regarding the panel data, the most common methodology is FMOLS, which Apergis and Payne introduced [26] for the study of six Central American countries for the period of 1971 to 2004, providing evidence of an inverted U-shaped EKC. The same conclusion was obtained by Liu et al. [31] referring to 10 newly industrialized countries for the period from 1971 to 2013, and Aruga (2019) [27] for a number of Asian Pacific countries.

2.2.2. EKC and Deforestation

Deforestation is a process that is increasing as a result of economic expansion. Therefore, interaction among deforestation and prosperity has become a subject of extended studies with significant efforts to be succinctly described below.

Cropper and Griffiths [32] valued deforestation not as an environmental degradation index, but as an environmental management measure. Specifically, this study involved the impact of population and income increase on the reduction of forest areas for 64 developing countries including those in Africa, Asia, and Latin America, which all feature huge forests and forest areas. According to these findings, the EKC was validated for countries in Latin America and Africa, where the per-capita GDP is lower than the estimated highest point of the curve ($5.420 and $4760 respectively in 1985 prices or at about $9100 and $7900 in 2001 prices. Therefore, the ECK did not reach the highest point, implying limitations in the validation of the EKC.

The constant growth of the global population and the adverse exploitation of natural sources to satisfy human needs, have led to deforestation along with conversion to agricultural lands.

The depletion of the world's forests in both tropical and temperate regions is causing considerable environmental problems that hamper sustainable economic development. In approaching the deforestation trend, some researchers argued that this activity might be slow or reverse, unveiling the validity of the EKC hypothesis. Nevertheless, the results and conclusions of studies investigating EKC contradict each other. Moreover, the relevant literature compared OECD countries with the non-OECD countries of Latin America, Asia, and Africa in determining the ways under which the various factors of economic growth, population, trade, urbanization, agricultural land conversion, and cereal yield impact deforestation rates [33]. These authors concluded that the OECD countries present an N-shaped curve, while for the African region, an income-based EKC pattern is validated. It is also noteworthy that the trade openness and the internationalization of the marketplace, along with widespread urbanization, are all impacting the regions in a different way, but only these countries have shown less deforestation attributed to higher cereal yields [33].

The validity and credibility of the EKC method to examine the deforestation process has attracted plentiful studies. Even though deforestation is widely studied, the relationship between economic development and deforestation remains questionable. In this context, a meta-analysis of selected EKC studies for deforestation included 69 studies, offering 547 estimations regarding the differentiation and vulnerability of EKC-reported results [34]. These authors investigated a range of choices—such as econometric strategy, measures of deforestation, geographical area, and the presence of control variables—on the probability of finding an EKC. It was argued that the validity and credibility of the EKC method is solidified, and the theoretical alternatives to the predominance of EKC as a method to examine the deforestation process could fade [34]. In deepening scientific understanding upon the functionality of EKC and in perceiving alternative means of explaining how capitalism and ecological disorganization (pollution) are interconnected under Marxist theory, Lynch [35] illustrated the limited ability of common EKC interpretation and attempted to implement a Marxian interpretation of the EKC. Under this context, Marxian analysis supports a reasonable interpretation for the inverted "U" shape of pollution where increased pollution is expected due to ongoing economic development. Furthermore, occasionally, the orthodox EKC may fail to fulfill EKC predictions in developing countries, thus implying that pollution rises in these areas. In the long term, pollution is reduced in poor nations, but it leads to a rise in pollution in the global context. However, as time passes, the production in developing economies is increasing, which leads in turn to an upward trend for pollution [35].

The main socio-environmental parameters that determine deforestation are the: property rights, the agricultural price index, the forest area, population, income, and timber prices [36]. Therefore, a plausible methodological tool to investigate the impact of deforestation-induced factors is the EKC. Under this research context, Esmaeili and Nasrnia [36] deployed the autoregressive distributed lag approach to yield the deforestation function, confirming the existence of an inverted U-shaped EKC for deforestation in Iran. Moreover, their research proved a linkage among deforestation and property rights, forest area, agricultural price index, and terms of trade. Additionally, the key aspects that support sustainable forest land uses and a reduction in the deforestation rate are the improvement of secure property rights and the environmental policies planning in Iran [36].

Regarding the roles of trade openness and agricultural productivity under the EKC, researchers have investigated these factors under the applicability of the EKC hypothesis in relation to emissions from the non-oil sector, such as agriculture (i.e., all GHG emissions from agriculture) among the sub-Saharan African countries [37]. Particularly, authors were focused on two indicators of environmental change, namely that of "rate of deforestation" and "all greenhouse gas emissions from agriculture" (Agri-GHG) in addressing whether the EKC hypothesis exists for both indicators applied and, secondly, to investigate the effects of macroeconomic and institutional variables on both of these previously considered indicators. Specifically, it is noteworthy that: (a) the EKC exists (such as an inverted "U"-shaped) only for all GHG emissions from agriculture; (b) agricultural production and trade openness significantly increase both of the applied environmental change indicators, and (c)

population growth significantly reduces Agri-GHG, while economic growth significantly increases the "rate of deforestation" in the region examined [37].

Zafeiriou et al. [38] studied the temporal environmental degradation agricultural income relationship for three ex-socialist newly entrants in the EU (Bulgaria, the Czech Republic, and Hungary), for which the EKC hypothesis was confirmed for the first two countries, while for Hungary, it was not confirmed upon the specific data employed.

Barbier and Burgess [39] focused on a study of deforestation estimation in tropics through the extension of cultivated areas. The sample involves countries of the tropics in Africa, Latin America, and Asia for the period from 1961 to 1994. According to their findings, the EKC was validated as a turning point, generating $8700.00 GDP per capita in 2003 prices. The model included a number of economic and social variables, such as GDP growth, the population increase rate, cereal production, land-use distribution, the exports of agricultural products, political corruption, and political stability, as well as property rights implementation. According to these findings, the expansion of agricultural area was positively affected by the population increase rate, while the agricultural land distribution, exports, and political stability positively impacted the dependent variable [39].

Similarly, Ehrhardt-Martinez et al. [40] utilized data for the time period from 1980 to 1995 and for the 74 least developed countries in Africa, Asia, and Latin America. These authors studied the validity of the ECK hypothesis and signified that the threshold value is estimated at $1150 in 1980 prices (or $2354.00 in 2003 prices).

In another study regarding deforestation, Lantz [41] used the annually deforestation rates in five Canadian regions for the time period of 1975 to 1999 as proxy for forest area deforestation in order to unveil different relationship patterns, including those between income–deforestation, population–deforestation, and technological improvements–deforestation, across time. According to these findings, the deforestation area was negatively related to income evolution, which was constantly decreasing. Therefore, the impacts on the forest destruction lead to higher incomes, and they are more efficiently reflected to the population [41].

Chiu [42] validated that deforestation was decreased in alignment with the income increase in more than 52 developing countries. Similarly, Farhani et al. [43] confirmed the validity of EKC for the relationship between income–environmental sustainability in South African and Middle Eastern countries. The study of Parajuli et al. (2019) [44], who studied the impact of forests, agricultural area, and energy consumption on the carbon emissions generated by 86 countries by developing the dynamic panel data approach, is also noteworthy.

2.2.3. EKC Methodological Issues

The validity of the environmental degradation–income per capita relationship has been subjected to plentiful studies within the last two decades, which is in alignment with the different indexes considered, and also with different linear and nonlinear methodologies applied [38,45–52]. The deterioration of the environmental problems caused by carbon emissions and especially GHG emissions, the identification of the impacts of climate change on global economy [53], and the concepts of eco-efficiency sustainability under the context of the global economy supported motives for these theoretically approached scientific fields. The prevailing econometric methodology upon the environmental degradation–economic growth relationship is either linear or nonlinear cointegration. The inverted U pattern of this relationship was initially introduced by Grossman and Krueger [47]. Esteve and Tamarit [54] outlined the most significant empirical studies, including those performed by Ozturk and Acaravci [55], Halicioglou [56], Soytas and Sari [57], and Soytas et al. [23], showing contradicting results with linearity to be the common feature among all of them. The existence of nonlinearities was initially studied by Esteve and Tamarit [54], noting that the implementation of the threshold cointegration validated the EKC hypothesis for Spain over a long-run reference period.

The present study, with the aid of the ARDL cointegration technique, investigated the relationship of environmental degradation–economic growth by using the: (a) carbon emissions generated due

to deforestation as an index for environmental degradation, and (b) GDP per capita as an index for economic growth for the new EU entrant country of Bulgaria. The use of this index for environmental degradation is the novel contribution of this study, comparing to existing literature.

## 3. Data Methodology

### *3.1. Data*

In the study, authors employed carbon emissions generated by deforestation as an index for environmental degradation for the time period from 1990 to 2015. On the other hand, as an index for economic growth, the GDP per capita is considered for the same time period. Both variables refer to the country of Bulgaria. The data employed were derived by the FAOSTAT database. The variables involve the carbon emissions equivalent in thousands of tonnes derived by deforestation for the 1990–2015 time period and per capita income as an index for economic growth.

The variables employed in the study are illustrated in the following Figures 1 and 2, respectively. Figure 1 depicts the evolution of GDP per capita for the time period between 1990–2015. Specifically, there is an evident sharply increasing trend over the period from 2004 to 2005, whereas in the last decade, the GDP per capita has been doubled, and has been stable ever since.

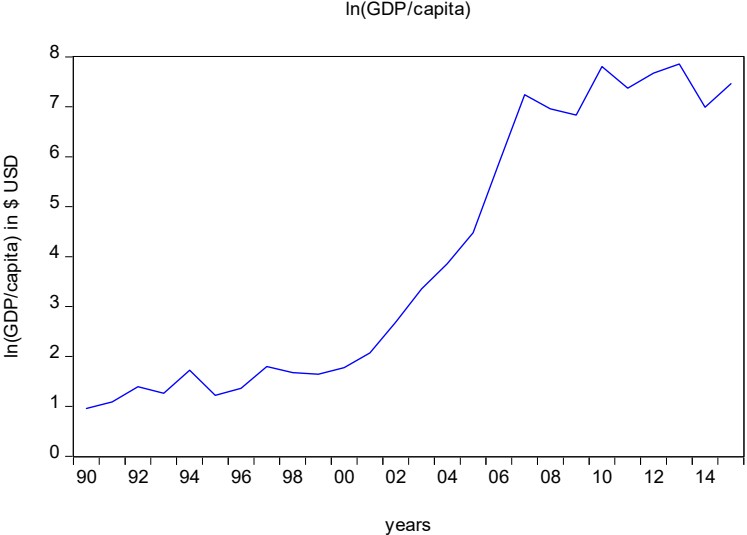

**Figure 1.** ln(GDP/capita) for Bulgaria. GDP: gross domestic product.

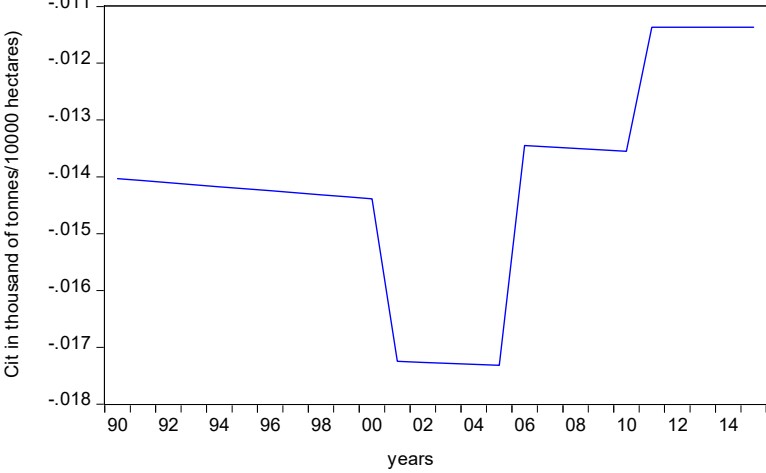

**Figure 2.** Evolution of carbon emissions generated by deforestation (1990–2015) for Bulgaria (ln(Cit).

Figure 2 shows the per capita GDP (in USD), with an upward trend and stability after the year 2010. Figure 2 shows the evolution of carbon emissions equivalent (in ln), although the pattern of its evolution cannot be defined with accuracy.

## 3.2. ARDL Methodology

The methodology employed is the ARDL bounds cointegration technique. The following ARDL model is used for the scope of this study:

$$\Delta LnC_t = \alpha_0 + \quad \alpha_1 T + \sum_{i=1}^{p-1} \alpha_{1i} \Delta lnC_{t-i} + \sum_{i=0}^{p-1} \alpha_{2i} \Delta lnGDP_{t-i} + \sum_{i=0}^{p-1} \alpha_{3i} \Delta lnGDP^2_{t-i}$$
$$+ \sum_{i=0}^{p-1} \alpha_{4i} \Delta lnGDP^3_{t-i} + \phi_1 lnC_{t-1} + \phi_2 lnGDP_{t-1} + \phi_3 lnGDP^2_{t-1} + \epsilon_t \tag{1}$$

where $C_t$ denotes carbon emissions generated by deforestation, *GDP* denotes gross added value per capita, T denotes the time trend, $\alpha_0$ the constant, $\Phi_{1,2,3}$ denotes the long-run coefficients, $\alpha_{1i}$, $\alpha_{2i}$, $\alpha_{3i}$, and $\alpha_{4i}$ denote the short run parameters, $\Delta$ denotes the first difference operator of the variable employed, and P denotes the lags determined by the employed lag length optimization criteria such as Akaike Information Criterion (AIC) and Schwarz Bayesian Criterion (SBC). The general model provided above is supplied under the condition that it suffers from stability problems; a modified model might be employed, since the time trend T or the constant coefficient should be omitted.

Equation (2) includes the long-term and the short-term parameters. Implicitly, $\varphi_1$, $\varphi_2$, $\varphi_3$, and $\varphi_4$ denote the long-term parameters, and therefore reject the null hypothesis; $\varphi_1 = \varphi_2 = \varphi_3 = \varphi_4 = 0$ (no cointegration) against the alternate (that is $\varphi_1$, $\varphi_2$, $\varphi_3 \neq 0$) confirms that the variables are cointegrated. The cointegration test is based on the computed F-statistic, being compared to the critical values provided for small size data by [58]. The potential results of this test are synopsized in Table 1.

**Table 1.** Results of F statistic test.

| Condition | Result |
|---|---|
| Fstatistic > UCB | Cointegration |
| Fstatistic ≤ LCB | No cointegration |
| LCB ≤ Fstatistic ≤ UCB | Uncertain (Result depends on the lagged error correction term for a long-run relationship) |

One of the advantages of the specific methodology involves its efficiency when implemented to variables that have different orders of integration, but this order should be less than two [59]. Therefore, prior to the estimation model and testing the existence of cointegration, it is a prerequisite to explore the order of integration for the variables used in the model. Implicitly, the authors examined the existence of a unit root in the behavior of GDP per capita and carbon emissions equivalent generated by deforestation. The unit root tests employed are the DF-GLS test and the augmented Dickey Fuller (ADF) with break point test. The major advantage of the last-mentioned test involves the simultaneous detection of potential structural breaks along with stationarity for the time series studied.

The break unit root test involves a two-step procedure. In the first step, the authors detrended the series with the appropriate trend and break variables, and also with the assistance of ordinary least squares (OLS), and in the second step, the existence of a unit root with the assistance of a modified Dickey–Fuller regression is implemented to the detrended series [60]. The initial model may involve non-trending or trending data with an intercept break or trend break. The results refer either to the trend or to the break specification (trend, intercept, or both).

Misspecifications are a potential problem of improper unit root test selection. For that reason and for the selection of an optimal method and model of the unit root test, this study used the one proposed by Shrestha and Chowdhury [61].

Having confirmed that the variables employed are not I(2) allowed the authors to implement the ARDL approach. The slightly modified Johansen cointegration technique—namely, the ARDL model—is free from residual serial correlation and endogeneity problems [62]. A number of advantages can be mentioned—including, among others, that the lags employed in the model are selected in a general-to-specific modeling framework [61–63]. Furthermore, the dynamic error correction model (ECM) through a simple linear transformation [64] integrates the short-run dynamics with the long-run equilibrium. Another serious advantage is related to the lack of problems resulting from non-stationary time-series data [63].

Prior to the model estimation and according to the methodology suggested by Pesaran et al. [65], the authors chose the lags for optimum model. The selection of the best model was based on the lowest prediction error. The next step confirmed the existence of a long-run relationship with the assistance of the bounds test and the error correction model.

The long-run relationship to be surveyed is the following for the ARDL (p,q,r) model:

$$lnC_{it} = \lambda_1 lnGDP + \lambda_2 lnGDP^2 + \lambda_3 lnGDP^3 \, u_t + \mu + \lambda 3D_t + \delta_t \tag{2}$$

where $lnC_{it}$ denotes the carbon emissions generated by deforestation for the country to be surveyed, *lnGDP* denotes the gross domestic product (GDP) per capita trend in logarithmic form, *lnGDP²* denotes the quadratic form of the GDP per capita, $lnGDP^3$ denotes the cubic form of the GDP per capita, and $D_t$ is a dummy variable that captures a single structural break. This structural break may be detected in the relationship among the variables studied for Bulgaria, in case that is statistically significant. In case the parameters $\lambda_1$ and $\lambda_2$ are found to be positive and negative respectively as well as statistically significant, the EKC is validated.

In the last step of the analysis, a sensitivity test was performed, through which parameter stability and goodness of fit is tested, which includes the cumulative sum of squares of recursive residuals (CUSUM test). The parameter stability is ensured in case the graphs mentioned above lie within the bounds [65,66].

The error correction model is provided by Equation (3):

$$(1-L)\begin{bmatrix} lnC_t \\ ln(GDP)_t \\ ln(GDP^2)_t \\ ln(GDP^3)_t \end{bmatrix} = \begin{bmatrix} \varphi_1 \\ \varphi_2 \\ \varphi_3 \\ \varphi_4 \end{bmatrix} + \sum_{i=1}^{p}(1-L)\begin{bmatrix} \alpha_{11i} & \alpha_{12i} & \alpha_{13i} & \alpha_{14i} \\ b_{21i} & b_{22i} & b_{23i} & b_{24i} \\ \delta_{31i} & \delta_{32i} & \delta_{33i} & \delta_{34i} \\ \varepsilon_{41i} & \varepsilon_{42i} & \varepsilon_{43i} & \varepsilon_{44i} \end{bmatrix} + \begin{bmatrix} \beta \\ \zeta \\ \gamma \\ \theta \end{bmatrix} ECM_{t-1} + \begin{bmatrix} \eta_{1i} \\ \eta_{2i} \\ \eta_{3i} \\ \eta_{4i} \end{bmatrix} \tag{3}$$

where (1-L) denotes the lag operator and $ECM_{t-1}$ denotes the lagged error correction term generated by the cointegrating equation, while the η terms are white noises. Short-run causality is confirmed in case the F statistic for the parameters of first differences is statistically significant, whereas the long-run causality is evident in case the lag of the error correction term with the assistance of t–statistics is reported as statistically significant.

## 4. Results

The results of the methodology are provided in the following Tables 2–7. In the first step of the analysis, two different unit roots tests were implemented, namely the DF-GLS unit root test and the DF break unit root test. The results of those tests are provided in Tables 2 and 3, respectively.

**Table 2.** Elliott–Rothenberg–Stock DF-GLS test.

| | *t*–Statistic | 5% Critical Value |
|---|---|---|
| lnCit | −1.11 | −1.955 |
| lnGDP | −1.148 | −1.955 |
| D(Cit) | −4.85 *** | −1.955 |
| D(GDP) | −4.39 *** | −1.955 |

*** Rejection of null hypothesis in 5% level of significance.

**Table 3.** Break unit root test; minimize Dickey–Fuller t-statistic.

| | *t*-Statistic | *p*-Value | Structural Break |
|---|---|---|---|
| lnCit | −2.964 | 0.71 | 2005 |
| lnGDP | −2.978 | 0.697 | 2002 |
| D(lnCit) | −6.927 *** | 0.01 | 2006 |
| D(GDP) | −4.803 *** | 0.012 | 2006 |

*** Rejection of null hypothesis in 5% level of significance; Critical value for 5% level of significance: −4.443649.

**Table 4.** F-test estimation result.

| F-Bounds Test | | Null Hypothesis: No Levels Relationship | | |
|---|---|---|---|---|
| Test Statistic | Value | Signif. | I(0) | I(1) |
| F-statistic | 7.39 | 10% | 2.37 | 3.2 |
| k | | 5% | 2.79 | 3.67 |
| | | 2.5% | 3.15 | 4.08 |
| | | 1% | 3.65 | 4.66 |

*** Rejection of null hypothesis in 5% level of significance.

**Table 5.** Long-run form.

| Variable | Coefficient | Std. Error | *t*-Statistic | Prob. |
|---|---|---|---|---|
| lnGDP | −13.39 *** | 0.456 | −29.31 | 0.0000 |
| lnGDP$^2$ | −0.143 *** | 0.018 | −7.620 | 0.0000 |
| lnGDP$^3$ | 2.65 *** | 0.190 | 13.96 | 0.0000 |

*** Rejection of null hypothesis in 5% level of significance.

**Table 6.** Error correction model.

| Variable | Coefficient | Std. Error | *t*-Statistic | Prob. |
|---|---|---|---|---|
| D(GDP$^3$) *** | −0.176 | 0.025144 | −6.452 | 0.0000 |
| D(GDP(-1)) | −0.0035 | 0.002349 | −1.487 | 0.23 |
| D(GDP$^2$) *** | 2.496 | 0.356405 | 6.583 | 0.00 |
| CointEq(-1) *** | −0.607 | 0.104571 | −5.899 | 0.00 |
| R-squared | 0.7156 | | | |

*** Rejection of null hypothesis in 5% level of significance.

**Table 7.** Autocorrelation and heteroscedasticity test of the model's residuals.

| | F-Statistic | *p*-Value |
|---|---|---|
| Breusch–Godfrey autocorrelation test | 0.28 | 0.75 |
| ARCH heteroscedasticity test | 1.118 | 0.268 |

According to these findings, the first unit root test confirmed that the variables used in the model—that is, deforestation as a proxy for environmental degradation and economic growth, respectively—are I(1), which is non-stationary in levels and stationary in first differences.

The second unit root test employed, which takes into consideration the existence of structural breaks, does also confirm that the variables are I(1), as evident in Table 3. Furthermore, this test provides the potential structural breaks of the time-series studied. Based on these, potential explanations for the structural breaks were identified. The year 2006 is a significant hallmark, since it coincides with the end of the privatization of the state-owned firms. This is significant for GDP per capita and deforestation due to a reduction in the foreign direct investments (FDI). In addition, regarding the year 2002, the mechanisms of coordination and management for the implementation of the strategy on structural funds have been refined. Finally, the year 2005 corresponds to when the Kyoto protocol was entered into force, specifically on 16 February 2005, which may adequately interpret the behavior of the carbon emissions equivalent generated by deforestation for the case of Bulgaria.

Having empirically confirmed that the time series studied are not I(2), the ARDL methodology was well deployed. The ARDL model selected based on the Akaike criterion, was ARDL (1,0,2,1). In the next step, the estimated F test suggests that the null hypothesis according to which no level relationships exist cannot be accepted, a result implying the existence of a long-run relationship among the variables studied, as evident in Table 4.

The next step in the analysis involves the estimation of the cointegrating relationship (long-run relationship) model based on the ARDL bounds test and is provided in Table 5.

The estimated error correction term that describes the speed of convergence to the steady state is the following: Cit − (−10.4642*GDP_CAPITA + 2.0064*(GDP)$^2$ − 0.1017*(GDP)$^3$ − 3.0495). The coefficients of the long-run relation are found to be statistically significant (for 10% level of significance), while the signs of the coefficients are as follows; $\lambda_1 < 0$, $\lambda_2 > 0$, and $\lambda_3 < 0$. Therefore, the signs validate the reversed N Kuznets curve pattern. This pattern implies that emissions would begin to rise again once a second income turning point is passed.

The estimation of the error correction model is provided in Table 6. The negative coefficients of $ECT_{t-1}$ are corroborating the short-term relationship in the model. The coefficient of the $ECT_{t-1}$ is indicative of the speed of convergence from short-term disequilibrium to the long-term equilibrium in the approximately 15.5 months in the linear ARDL model. The negative (positive) coefficients of $DGDP_{t-1}$ ($DGDP_{t-12}$) do not confirm the existence of the EKC hypothesis (inverted U pattern) with a one-year lag in the model. Subsequently, the inversed N Kuznets curve is also fully validated for the case of Bulgaria in the short run.

In the short term, it is shown that the statistical significance of the cubic form implies the validity of the N Kuznets curve pattern (not inverted N Kuznets, contrary to the long term). Regarding the diagnostic tests as observed in Table 7, we conducted the Breusch Godfrey autocorrelation test and ARCH heteroscedasticity of the estimated model residuals.

The last step in the analysis involves the study of the parameter's stability with the Cumulative d CUSUM square (CUSUMsq) tests, the results of which are illustrated in Figure 3. Specifically, in Figure 3, the particular plot lies within the critical bounds at a 5% significance level, which indicates that the estimated model is stable in the research period.

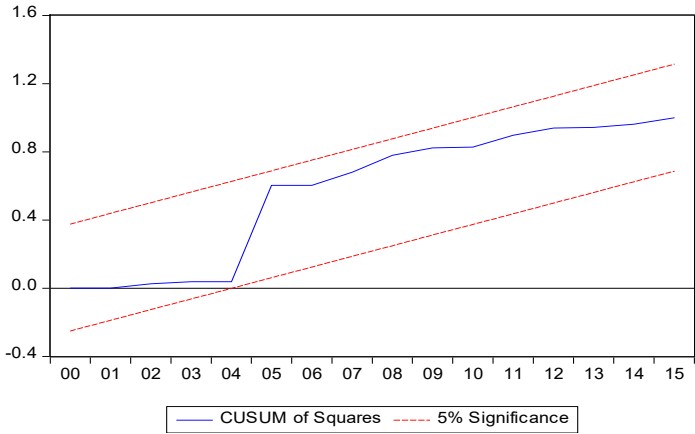

**Figure 3.** Illustration of cumulative sum of squares of recursive residuals (CUSUM) of squares stability test.

Furthermore, the parameter stability CUSUM of squares and the diagnostic tests provided in Table 7 confirm the robustness of the linear ARDL estimates test.

## 5. Conclusions

Among EU members, Bulgaria was selected as one of the newly entrant economies, having a high growth rate and specific condition in the deforestation process. This study investigated the validity of EKC in alignment with the ARDL approach of carbon emissions generated due to deforestation (in thousands of tonnes per 1000 ha of deforested ex forest land) as an index of environmental degradation, and the per capita GDP (in thousand of USD dollars) as an index for income. According to these findings, a reverted N Kuznets curve pattern was validated, which was a result implying two different income thresholds for change in the behavior of carbon emissions. This result was validated not only for the long-term period, but also for the N pattern relationship for the short term. Having confirmed the validity of the opposite N-shaped EKC implies that the economic growth initially will improve environmental quality to a certain income level, where the relationship will be positive before it once again becomes negative. This finding is not only interesting, but also challenging to interpret. Possibly, it could be a consequence of the initial environmentally-friendly attitude of the Bulgarian economy and its limitations toward deforestation as well as the efforts aiming to afforestation, compensating for the increased emissions caused by the scale effect. The results were based on carbon emissions generated by deforestation, and these may well efficiently and indirectly describe the forest land situation in Bulgaria, and contribute effectively to the design and implementation of environmental policies that are capable of eliminating the deforestation problem in Bulgaria.

The process of afforestation could provide a solution to the problem of deforestation—namely, a reduction in GHG emissions and an increased absorption of environmental pollutants. In addition, policies and measures design and implementation should promote the efficiency of human activities in order to ensure that the economic losses attributed to the limited exploitation of forest resources are limited.

The state's efforts for afforestation (a plausible explanation for the findings) were outperformed by the illegal logging, and the low wages of the civil servants in the forest service may also lead to a higher quantity of carbon emissions generated by deforestation. Therefore, initiatives should be taken for the counteraction of these behaviors that can be achieved by seminars, in order for residents to be better informed, and a provision of motivation for them to pursue a limitation in deforestation. This motivation could include, among others, economic incentives for the conversion of forestland to agricultural uses such as taxes or subsidies. Furthermore, clearly defined and enforced property rights to timberlands could also provide an effective solution to the limitation of the problem.

To synopsize the novel institutional measures that should be taken, we aim to limit the degradation of forest resources, which in turn will bring about a decrease in the carbon emissions generated by deforestation. Furthermore, the smooth forest land use with alternative methodologies may well lead to quality environmental improvement and the prosperity of rural and urban surrounding areas.

The specific conditions that dominate in Bulgaria despite the country complying with the agri-environmental measures adopted by the EU may also interpret the opposite N pattern of environmental performance–GDP per capita relationship, necessitating a more insightful study regarding the formation of the variables and the evolution of this relationship. For that reason, the findings of this study contradict the findings of Zambrano-Monserrate et al. [67] that confirmed the validity of EKC for other five European countries with the same methodology for a longer time horizon. Based on the aforementioned results, the environmental policies mainly for the case of Bulgaria should be directed to the expansion of forest land, since forestry production and agricultural exports may increase jointly, allowing a significant progress for environmental protection inside the continent.

The study also unveiled a remarkable conclusion regarding the shape of the EKC, thus suggesting research intensification in order for the pollution–income relationship to be identified. The relationship may well be studied in alignment with an alternative methodology while considering other possible shapes than those already examined and expected in other EKC core studies. It is important to further investigate the relationship between income and environmental degradation in order to combat climate change and reach sustainable economic development.

Conclusively, regarding suggestions for the future research, the implementation of a different nonlinear ARDL methodology on available data could provide more concise and accurate results, as well as the implementation of panel data analysis to support researchers with more general results regarding the agro-environmental EU policy for more countries.

**Author Contributions:** S.T. organized the study, conducted the design of the study, and analyzed the data. E.Z. deployed the analysis, the interpretation of results, as well as the revision of the study. E.Z. and G.K. modified and smoothed the written English. G.K. and G.A. deployed the literature review. S.T., E.Z., G.K., and G.A. formulated and finalized the conclusions of the study.

**Funding:** This research received no external funding.

**Conflicts of Interest:** The authors declare no conflicts of interest.

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
