# Peer review of "Revising the Environmental Kuznets Curve for Deforestation: An Empirical Study for Bulgaria"

_sustainability, doi:10.3390/su11164364_

Round 1
Reviewer 1 Report
Major comments:
Overall, an extensive revision must be done to meet the standard of the journal. It needs to explain backgrounds of the situation of the deforestation in Bulgaria and why using the EKC adds value to the relevant field. Second, the methods and results sections are written in a poor manner. The estimation results seemed to be using a cubic EKC but the methods showed a quadratic EKC. Conclusion section was also not written in a concrete way and needs to be rewritten. Also not much discussions were provided regarding the estimated results.
Minor comments:
1. p2, ln42-43. Please clarify whether the Gross Added Value shown in the table is global value or local value of some region or a country. Since the study is focusing on Bulgaria the authors should explain the background of the situation of the deforestation in Bulgaria or Eastern Europe.
2. p2, Table 1. please include the unit of the values.
3. p3, ln71-73, "The present manuscript tries to..." sentence. The reasoning of pursuing the objective with EKC is weak so more discussions should be made before bringing this sentence. The paper should argue why the paper use the EKC and which variables will be used to test the EKC in this study. They should explain especially why they used the EKC to deal with the deforestation issue. I think some of the discussions made regarding the validity of EKC in section 2 should be moved before this sentence.
4. p3, 2. Existing Literature. I feel this section should be more organized. 2.1 should only focus on explaining the EKC and they should create 2.2 to discuss previous studies applying EKC as mentioned in the next comment. As explained in the previous comment, I feel some of the discussion here should be moved to the Introduction to explain why EKC method would be valid for this study.
5. p3, ln91, [15-23]. These literature should be explained in a separate section combined with other literature presented in p5-6. I recommend the authors to create "2.2 Previous literature" section and discuss what types of studies have been done using the EKC combining those studies shown in p5-6. To contribute to the journal, I think it would be nice also to refer to the recent articles published in Sustainability using the Kuznets curve like the followings:
Sustainability 2019, 11(12), 3295; https://doi.org/10.3390/su11123295
Sustainability 2019, 11(9), 2688; https://doi.org/10.3390/su11092688
Sustainability 2019, 11(8), 2395; https://doi.org/10.3390/su11082395
6. p3, ln101-103. I don't see the closing parenthesis.
7. p3, ln108, Grossman και Krueger. "και" needs to be written in English.
8. p4-5. I feel this part needs to be rearranged and organized. Some of the literature might be better put in the Introduction. It's also better to separate into studies not related to deforestation and those using EKC for deforestation. Finally, it should be discussed what different results can be drawn from this study compared to those studies using EKC for deforestation.
9. p5, 3.1 Previous research. This whole section should be moved to Section 2.
10. p6-8, 3.2 Research Area. This whole section should be moved to 1. Introduction. Then, the authors should be able to show why pursuing the deforestation issue for Bulgaria using the EKC is important in the Introduction section.
10. p8, Figure 2. It should also have in the caption that the figure is for "Bulgaria"
11. p7-p9. After moving sections 3.1 and 3.2 to other parts of the manuscript, Section 3.3 and 3.4 can be now combined as one section as the "3. Methodology" section.
12. It should mention that the data used is for "Bulgaria."
13. p8, ln306-307. The "3" of "λ3" in equation (1) should be a subscript.  
The results of the paper suggest the cubic form of the EKC was analyzed. Does not the equation need to include the cubic lnGDP?
14. p8 equations (2) and (3) these equations also do not need to include the cubic lnGDP?
15. p9 equation (4). Here too also do not need to include the cubic lnGDP?
16. p10, ln382, "In the following tables are..." sentence. The English of this sentence is awkward and needs to be rewritten.
17. p10, ln385-387, "In the next tables 3 and 4 and..." sentence. This sentence too is poorly written and needs revision.
18. p10-11, Tables 3-7. The decimal place ordering must of consistent in all these tables. The authors should put * marks next to coefficients that are significant.
19. p.12, ln432. More explanation should be provided regarding the results of figure 3.
20. p10-15. This section is in general written in a very poor manner compared to the other sections. It should provide more detailed explanations about the results and implications that can be drawn from the estimations. Furthermore, I did not see any explanations for the reason of the outcomes. For instance, the authors should explain potential reasons why the EKC became a reversed N ECK pattern.
21. Conclusions section. This whole section needs to be rewritten because the implications driven from the results are not clearly explained how they are connected to the estimated results. Furthermore, they should provide more concrete policies that might derive afforestation in Bulgaria.
Author Response
answer to the reviewer 1

Reviewer 2 Report
The paper has a novel and interesting subject but there are needed some major improvement in order to have a publishable material. I will explain some modifications that may improve the status of the paper.
Line 30-31 – Need a reference.
Line 31-33 – It is a heavy speech, please revise.
Line 30-36 – These phrases could be rephrased and written in a more specific manner. It is hard to understand.
Table 1 – This table is more suitable on the results section and you only mention an idea in the introduction section (with reference).
Line 46 – Where is the first issue discussed? You have pass to “another issue”.
Line 46-48 – You need a reference for this statement.
Line 77 – You explain here some aspects that are found in the existing literature, as a meta-analysis research, but you must diminish this section “Environmental Kuznets Curve”. There are a lot of ideas that are more suitable on the section “material and methods”.
Line 221-233- You can add this section to the end of the section of “Environmental Kuznets Curve”.
The “Results” section is so poorly described, so, you must improve this aspect. It is so poorly described.
Author Response
Answers to the reviewer 2

Reviewer 3 Report
Paper: Revising the Environmental Kuznets Curve for deforestation: An empirical study for Bulgaria
The document looks for the existence of an environmental Kuznets curve in Bulgaria. The research presents interesting results, however, it needs to be strengthened before being accepted in "Sustainability". Below my comments:
1) Table 1 in the introduction is not appropriate. I suggest showing the data through a graph.
2) In the introduction, the authors can mention the most important results without giving so much detail. They should also mention the methodology that they will use. At the end of this section, it is important to mention the structure of the document.
3) Section 2 should be called simply "literature review" and they should remove the subtitle of line 77.
4) Delete lines 211-214.
5) In the methodological section, why do the authors speak again about previous research? I recommend eliminating this information from previous studies or to incorporate it into another section of the document.
6) In the methodological section, the authors should focus strictly on the description of the method used (ARDL), as well as the description of the variables and their source. On this, I suggest creating a table with a detailed description of the variables and their abbreviations.
7) In line 402, the authors mention "Long-term relationship", but then they continue with the short-term equation What is happening? Where are the long-term results?
8) The conclusions section is very weak. The authors should discuss in greater depth their results and compare them with other studies. They must also mention the limitations of their research.
9) The authors should check that all references have been cited in the text. Some of them are not located in the text.
10) Finally, the entire document needs to be reviewed and edited by a native English speaker.
Author Response
Answers to the reviewer 3

Round 2
Reviewer 1 Report
The authors still did not clearly explain why investigating the EKC for deforestation is important especially for Bulgaria. I think they need to add sentences to connect the situation of the economy, deforestation, and CO2 emission issue in Bulgaria and explain why the problem will be best explained by the EKC. The authors replied that “All the suggested references have been embodied to the text of the submitted revised manuscript,” but none of the papers below have been included in the revised manuscript.Sustainability 2019, 11(12), 3295; https://doi.org/10.3390/su11123295
Sustainability 2019, 11(9), 2688; https://doi.org/10.3390/su11092688
Sustainability 2019, 11(8), 2395; https://doi.org/10.3390/su11082395
Still the decimal place ordering in the tables is not consistent. I also suggest the authors to minimize the order of the decimal place. In p.7 Figures 1 and 2. The units in the vertical axis are missing. p.8-9. The numbers for the equations must be revised. p.9 equation (3). The cubic form of the GDP must be placed in the vector. The conclusion section is still not well-organized. The first paragraph of the conclusion is too long. The sentence “The results found due to lack of data for deforestation area– may well describe efficiently the forestland situation in Bulgaria contributing effectively to the design and implementation of environmental policies capable to eliminate deforestation problem in Bulgaria.” does not make sense so consider rewriting. The paper still contains quite a few typos and grammatical errors so the English must be proof edited by a professional English editor.Author Response
|
Reviewer 1 |
|
|
Comments |
Answers |
|
The authors still did not clearly explain why investigating the EKC for deforestation is important especially for Bulgaria. I think they need to add sentences to connect the situation of the economy, deforestation, and CO2 emission issue in Bulgaria and explain why the problem will be best explained by the EKC. |
done |
|
The quadratic EKC in equation 3. |
Addressed |
|
Still the decimal place ordering in the tables is not consistent. I also suggest the authors to minimize the order of the decimal place |
Done |
|
Conclusion section was also not written in a concrete way and needs to be rewritten |
The conclusion section was rewritten and corrected. |
|
p.7 Figures 1 and 2. The units in the vertical axis are missing. p.8-9. The numbers for the equations must be revised. p.9 equation (3). The cubic form of the GDP must be placed in the vector. |
Done |
|
All the suggested references have been embodied to the text of the submitted revised manuscript,” but none of the papers below have been included in the revised manuscript.
Sustainability 2019, 11(12), 3295; https://doi.org/10.3390/su11123295
Sustainability 2019, 11(9), 2688; https://doi.org/10.3390/su11092688
Sustainability 2019, 11(8), 2395; https://doi.org/10.3390/su11082395 |
Done |
|
“The results found due to lack of data for deforestation area– may well describe efficiently the forestland situation in Bulgaria contributing effectively to the design and implementation of environmental policies capable to eliminate deforestation problem in Bulgaria.” does not make sense so consider rewriting |
done |
|
The paper still contains quite a few typos and grammatical errors so the English must be proof edited by a professional English editor. |
Thorough English editing |

Reviewer 3 Report
No additional questions.
Author Response
All revisions have been done